# Assessing Burden, Anxiety, Depression, and Quality of Life among Caregivers of Hemodialysis Patients in Indonesia: A Cross-Sectional Study

**DOI:** 10.3390/ijerph19084544

**Published:** 2022-04-09

**Authors:** Theresia Maria Toji Pio, Junaidi Budi Prihanto, Yasmin Jahan, Naoki Hirose, Kana Kazawa, Michiko Moriyama

**Affiliations:** 1Graduate School of Biomedical and Health Science, Hiroshima University, Kasumi 1-2-3, Minami-Ku, Hiroshima 734-8553, Japan; tmtp.toji4547@gmail.com (T.M.T.P.); or d181423@hiroshima-u.ac.jp (J.B.P.); dr.yasminjahan@gmail.com (Y.J.); naoki-hirose@hiroshima-u.ac.jp (N.H.); kkazawa@hiroshima-u.ac.jp (K.K.); 2Nakajima Tsuchiya Clinic Hemodialysis Center, Hiroshima 730-0811, Japan; 3Department of Sport Education, Faculty of Sport Science, State University of Surabaya, Surabaya 60213, Indonesia

**Keywords:** anxiety, burden, caregivers, chronic kidney disease, depression, hemodialysis patient, quality of life

## Abstract

Family caregivers endure the burden of caring for patients receiving hemodialysis, which can affect their psychological status and may disrupt the care process. This study aims to assess the level of burden, anxiety, depression, and quality of life (QOL) among family caregivers, to investigate the influence of caregivers’ sociodemographic factors and patients’ clinical conditions on the level of burden, and investigate how burden affects anxiety/depression and QOL. A descriptive, cross-sectional study was conducted from September to October 2020. A total of 104 caregivers with a mean age of 44.4 ± 12.7 years (63.5% women) in the hemodialysis department of a hospital in Indonesia were examined. Zarit Burden Interview (ZBI), Hospital Anxiety and Depression Scale, and WHOQOL-BREF were used. Descriptive analysis was conducted to assess the level of psychological status, and multiple regression analysis and path analysis were performed to evaluate the association among all factors. As result, regarding burden, 10.2% had a moderate-to-severe burden, and in terms of anxiety and depression, 25% and 9.6% showed abnormal cases; the mean scores of 4 domains of QOL were about 60 points. Burden and anxiety/depression were significantly negatively associated with all domains of QOL (*p* < 0.001); however, sociodemographic and clinical factors were not associated with any of the QOL domains. The path analysis results showed that burden positively correlated with anxiety/depression and negatively correlated with QOL via anxiety/depression. In conclusion, family caregivers’ QOL was found to be indirectly influenced by burden through anxiety/depression. Further evaluation of clinical significance and implications for caregiver’s lifestyle regarding anxiety and depression, which result in caregiver burden, is warranted.

## 1. Introduction

Globally, the prevalence of all-stage chronic kidney disease (CKD) was recorded at 9.1% (697.5 million cases) in 2017, and the all-age CKD prevalence was increased by 29.3% between 1990 and 2017 [1]. In a developing country such as Indonesia, a rising incidence of CKD [2] and end-stage renal disease (ESRD) have been reported, and hemodialysis is considered the major therapy for ESRD [3]. In Indonesia, the number of new cases of patients with ESRD undergoing hemodialysis has increased almost four times from 2014 to 2018 [4]. The reasons may be due to the increasing trends of lifestyle diseases, especially obesity, hypertension, diabetes, lack of health literacy, and failure to visit hospitals at the early stages of CKD [5]. However, although hemodialysis prevents the death of patients with ESRD, it causes significant changes in their lifestyles. Hemodialysis causes physical and financial burdens, as well as psychosocial challenges to patients and their caregivers [6]. Evidence suggests the role of caregiving for patients receiving hemodialysis is associated with high burden, depression, anxiety, and impaired quality of life (QOL) [7]. 

The degree of burden, anxiety, depression, and QOL of caregivers are affected by multiple factors. It has been reported that the caregiver’s burden was significantly correlated with the age of the caregiver [8], female spouse [8], single [9], a lower education level [10], lower socioeconomic status [11], number of hours spent in caregiving [12], and giving a long term of caregiving process [12].

Patients’ sociodemographic and clinical conditions also affected caregivers’ physical and psychological burdens—namely, being male [13], having a lower income [13], having multiple comorbidities [10], and the level of patient’s self-care ability [14], which further reduce their QOL [6]. Regarding the social impact of caregiving, recent studies have revealed deteriorated family relationships [15], stress [15], social isolation [16], lack of confidence [17], fatigue [18], lack of independence [19], and financial constraints [20] may affect the caregivers’ physical, social, and emotional well-being, thus reducing their QOL [21]. 

Additionally, the care burden affects caregivers’ QOL and may result in reduced care provision and deteriorating conditions for patients with chronic illness. The deterioration of a patient’s condition can increase the care burden and cause a vicious cycle, and if timely intervention is not carried out, it may lead to the gradual exhaustion of the caregiver [22]. Therefore, timely identification of these burdens in caregivers plays a decisive role in promoting their physical and mental health [23]. However, to date, no studies have been conducted related to the caregiver’s QOL and its relation to burden, anxiety, and depression. Considering this issue, this study aimed to identify the factors associated with caregivers’ QOL and its relation to burden, anxiety, and depression regarding patients receiving hemodialysis. 

The purposes of this study were (1) to assess the level of burden, anxiety, depression, and QOL among family caregivers of patients receiving hemodialysis in Indonesia, (2) to investigate the influence of family caregivers’ sociodemographic factors and patients’ clinical conditions on the level of burden, and (3) to investigate how burden affects anxiety/depression and QOL.

This study hypothesized that (1) caregivers’ sociodemographic factors and patients’ clinical conditions affect the level of burden, (2) burden affects QOL, and (3) burden affects anxiety/depression in relation to QOL. The research framework was based on research findings and the above hypotheses (Figure 1).

## 2. Materials and Methods

### 2.1. Study Design, Setting, and Sample

A descriptive, cross-sectional study was conducted in central Jakarta, the Capital city of Indonesia, between September and October 2020. There are 22 hemodialysis facilities in Jakarta (Faskes, 2020). The Sint Carolus Hospital (private) was chosen because it is a hospital with 192 beds, having one of the largest outpatient hemodialysis units for 150 patients in 2019. Participants were unpaid family caregivers who take care of patients with CKD having hemodialysis on an outpatient basis for at least three months at the hospital. The inclusion criteria were 18 years old and above (irrespective of sex), main caregivers who provided assistance with daily activities, including hands-on care, care coordination, and accompanying patients during a hemodialysis treatment, and who agreed to participate in the study. Exclusion criteria were caregivers who presented difficulty in understanding the questionnaires and refused to participate in this study. The sample size was calculated by G*Power 3.1.9.2. Multiple linear regression and bivariate analyses were chosen. Based on a medium effect size 0.15, α (alpha) error probability <0.05, and power (1 − *β* error probability) 0.80, sample size became 89. Assuming the dropout, the number of respondents was increased to 20%. Therefore, the total number of participants in this study was 107.

### 2.2. Procedure

A convenience sampling procedure was used for this study in the selected above-mentioned hospital. A detailed description of the study, benefits, confidentiality, and the informed consent procedures was explained during the initial contact with the prospective participant before their participation. An anonymous, self-completed online questionnaire was returned and considered as consent to participation in the study. The research assistant explained the purpose and consent procedure of this study to the caregivers individually in the family waiting room of the hemodialysis ward. When the caregivers agreed to participate, the Google Form questionnaire was stored on the iPad of participants. To maintain privacy, no identifying personal information was collected. When participants did not understand the question or sentence in the questionnaire, they were allowed to ask the researcher’s assistant. After completing the questionnaire, participants submitted their answers voluntarily to the Google Form site.

### 2.3. Measures

#### 2.3.1. Sociodemographic Data

The sociodemographic data were divided into two categories: one for caregivers and another for patients. The caregiver’s sociodemographic factors data included gender, age, level of education (junior high school or lower, senior high school, diploma/bachelor, master or higher), marital status (single, married, widow), employment status (unemployed, employee/self-employed), monthly income (low, lower-middle, upper-middle, high), relationship to the patient (parent, spouse, child, siblings, another relative, caregiver). Data on patients’ sociodemographic factors and clinical conditions included gender, age, comorbidities (chronic diseases such as cardiovascular, diabetes, etc.), hemodialysis duration (months and years), number of dialysis sessions per week, hours of dialysis per session, and patient’s ability to accomplish their daily tasks (independent, half-dependent, fully dependent).

#### 2.3.2. Burden by Zarit Burden Interview (ZBI)

The caregiving burden was measured by ZBI [24]. It consists of five domains: burden in the relationship, emotional well-being, social and family life, finances, and loss of control over life. The 22-item ZBI has a 5-point Likert scale ranging from 0 (rarely) to 4 (nearly always) except for the final item, which has 5 ordered intensity-related response categories (0 = not at all; 4 = extremely). The total score ranges from 0 to 88, with a higher score indicating a heavier burden. A score of 0–20 indicates little or no burden; 21–40 means a mild-to-moderate burden; 41–60 means a moderate-to-severe burden; 61–88 means a severe burden. The Indonesian version of ZBI was validated and found to be reliable, with Cronbach’s alpha value of 0.91 [25].

#### 2.3.3. Depression and Anxiety by the Hospital Anxiety and Depression Scale (HADS)

HADS was developed to measure the symptoms of anxiety (HADS-A) and depression (HADS-D) for both dimensional and categorical aspects [26]. The questionnaire comprised 7 questions for anxiety and 7 questions for depression. Although the anxiety and depression questions are interspersed within the questionnaire, they are scored separately. Each item is scored on a 4-point Likert scale (0 = not at all to 3 = nearly all the time); thus, each subscale can range from 0 to 21. Scores are interpreted as non-cases (0–7), mild (8–10), moderate (11–14), or severe (15–21) symptoms. In Indonesia, Rudy et al. tested the Indonesian version of HADS for reliability. The Kappa coefficient for inter-rater agreement of HADS for the anxiety subscale was 0.706 and for the depression subscale, it was 0.681 [27]. 

#### 2.3.4. Quality of Life Measurement Using the World Health Organization Quality of Life: Brief Version (WHOQOL-BREF)

WHOQOL-BREF was used to assess QOL in four domains—namely, physical health, psychological, social relationships, and environmental health domains. The first two items are about overall QOL and general health. Each item is scored on a Likert scale from 1 to 5, where 1 represents “very dissatisfied/very poor”, and 5 represents “very satisfied”. The score is then transformed into a linear scale between 0 and 100 scale, where a score of 0 is the least favorable and a 100 is the most favorable [28]. The Indonesian version of WHOQOL-BREF has been proven as a valid and reliable questionnaire, and all of the domains met the reliability criteria (Cronbach α was ≥0.6). Therefore, the Indonesian version of the WHOQOL-BREF was used for this study [29,30]. 

### 2.4. Data Analysis

For statistical analysis, the statistical software SPSS and AMOS (Ver. 25, IBM, Armonk, NY, USA) were used, with the significance level set at under 5%. The participants’ sociodemographic factors and clinical conditions, ZBI, HADS, and WHOQOL-BREF results were described with mean and standard deviation (SD), for continuous variables and percentage for categorical variables. Prior to the calculation, to confirm the reliability of each scale and correlations in HADS subscales, Cronbach’s α coefficients were calculated. To examine the correlation between sociodemographic factors of caregivers and clinical conditions of patients, as independent variables, on the level of burden, as a dependent variable, a simple linear regression was used for analysis. For this purpose, 8 variables were recategorized from 14 variables, which are shown in Table 1. Next, multiple regression analyses (enter method) were performed to evaluate the association between burden (ZBI) or anxiety and depression (HADS) as independent variables and each domain of QOL as dependent variables. Sociodemographic variables and clinical conditions were added as confounding factors. Furthermore, based on the results, models were created for the overall relationships among all factors and were examined to observe to what extent they matched our data using the path analysis method. 

### 2.5. Ethics Statement

The study was approved by the author’s University’s Ethics Review Committee (Number.13/UN25.1.14/EPK/2020). The study follows the principles put forth by the Council for International Organizations of Medical Sciences (CIOMS) regarding the guidelines for health-related research involving humans. 

## 3. Results

### 3.1. Sociodemographic Factors of Caregivers and Clinical Condition of the Patients

The questionnaire was administered to 108 caregivers, and 4 who did not match the inclusion criteria were excluded (the valid response rate was 96.3%). The sociodemographic characteristics of 104 participants are listed in Table 1. The results reveal that most of the participants were females, married, employed, and had spouses. Among them, 72.1% had completed their diploma/bachelor or higher levels of education. Moreover, 35.6% had high income, whereas 21.2% (*n* = 22) had lower-middle-income status (around USD 352, in Indonesian currency Rp. 5,000,000/month). 

There were more male patients, and their topmost comorbid condition was hypertension, followed by diabetes. In the yearly hemodialysis proportion, a total of 67.3% of patients received ≤3 years of dialysis. As for the frequency of dialysis, most of them received hemodialysis 2 times/week session and 5 h per session. As regards the ability to perform their daily tasks, 55.8% of patients were half- or fully dependent, whereas 44.2% were independent (Table 1).

### 3.2. Levels of Burden Anxiety, Depression, and QOL

The reliability of each scale showed enough stability except HADS-D (ZBI: α = 0.911, HADS-A α = 0.878, HADS-D α = 0.695, WHOQOL-BREF: α = 0.909). Mean (SD) scores of all measurements are described in Table 2. Regarding burden, 10.2% had moderate-to-severe levels of burden, and only one caregiver had a severe level of burden. As regards anxiety, 13.5% showed borderline cases, and 25% showed abnormal cases. Regarding depression, 20.2% were borderline cases, and 9.6% were abnormal cases. Mean scores of 4 domains of QOL were around 60 points and over.

### 3.3. Characteristics of Caregivers and Patients Influence the Burden Level

The influences of sociodemographic and clinical factors on burden are shown in Table 3. Almost 90% of patients who received hemodialysis twice per week and 5 h per session were not included in these variables. As a result, caregivers’ characteristics or patients’ clinical conditions had no statistically significant influence on the burden level (all, *p* ≥ 0.05). 

### 3.4. Association between Burden, Anxiety, and Depression with QOL

A Pearson correlation was calculated between HADS-A and HADS-D, and the result showed a moderately high correlation (r = 0.677, *p* < 0.01). Additionally, factor analysis was checked if all items were categorized into HADS-A and HADS-D by the method of maximum likelihood (Varimax rotation). As result, HADS were divided into two subscales, but items categorized as anxiety and depression were all mixed. Therefore, HADS was considered as one scale.

The results showed that burden (Table 4) and HADS (Table 5) were significantly negatively associated with all domains of QOL (all, *p* < 0.001), which indicates when the burden or HADS (anxiety and/or depression) became severe, QOL (all domains) decreased. However, sociodemographic and clinical factors were not associated with any of the QOL domains. 

### 3.5. Path Analysis for Burden, Anxiety, and Depression with QOL

Based on our hypothetical model (Figure 1) and the results of multiple regression analysis, a model for the causal relevance was created in each QOL domain. The final models that demonstrated the best fit with the data as results of path analysis showed that burden positively correlated with HADS, and negatively correlated with QOL (each domain) via HADS. The fitness values of models in all QOL domains are listed in Table 6. All models showed good fitness; however, the burden variable can explain only 45% of HADS, and HADS can explain only 19% (overall health) to 42% (psychological QOL) alone (Appendix A).

## 4. Discussion

To date, this is the first exploratory study on burden, anxiety, depression, and QOL among family caregivers of hemodialysis patients in the Indonesian context. This study was conducted to assess the level of burden of the patient’s caregiver receiving hemodialysis in Indonesia. It was assumed that the majority of the caregivers would report moderate-to-severe burdens; however, this study found more than half of the caregivers had little or no burden. This is similar to a study conducted in Turkey, which found a low burden (45.6%) [6], and unexpectedly, so indicated the results from Nepal (49.4%) [31] and Vietnam (80.9%) [8]. Interestingly, developed countries such as Japan, Sweden, the UK, and the USA, where nursing care at the bedside and long-term care are well established, reported a high burden (30–47%) [32]. The level of burden experienced by caregivers can be influenced by many factors, including governmental and non-governmental support of caregivers and the dominant culture of society. Regarding this, and based on existing evidence [13,33], burdens in developing countries also appear to be higher. 

This study hypothesized that sociodemographic factors would affect the burden of caregivers; however, no significant association was observed with these aspects. This study showed that more than half of the caregivers were younger and aged less than 40 years. The majority of caregivers had completed their diploma/bachelor’s or higher levels of education, which was not consistent with the Indonesian educational background [34]. Additionally, about half of the caregivers worked, and more than half were from upper-middle-income status, which was also different from Indonesian economic status (monthly income Rp.1,400,000 to 1,900,000/month (USD 100–140/month)) [35]. The reason may be because participants in this study were from a hospital located in the capital city, where patients belong to middle-to-high income status. This could be the reason behind a high percentage of caregivers being educated and having a high monthly income. Therefore, this study’s population may have a better sociodemographic background, compared with the general Indonesian population. However, several factors such as the patients being independent [36,37], having insurance coverage [37], patients and caregivers being young [38], and religious beliefs [12] could possibly explain the low levels of burden found in this study. 

Further, identifying a caregiver’s psychological status is clinically important, as untreated anxiety may lead to depression [39], and in the long run, poor caregiver mental health is associated with a patient mortality rate [40]. The findings of this study showed that one-third of the caregivers experienced borderline and abnormal cases of anxiety and depression. Several studies have found that the incidence of mental illness of the main caregivers of chronic disease patients is equal to or greater than that of the chronic disease patients, which leads to a decrease in QOL [9,41]. This study’s hypotheses are partially supported by the results indicating the QOL of hemodialysis caregivers is poorer, compared with the general population [15], which supported this study’s hypothesis that burden, anxiety, and depression are related to QOL. However, no direct, significant association was observed between burden and QOL. We assumed that WHOQOL includes items such as physical pain, self-esteem, activities of daily living, personal relations, and home environment, which might not be affected by the caregiving burden.

Nevertheless, patients on dialysis around the world may have unpaid caregivers, who appeared to have more comorbidity, worse QOL, more depression, and to be lower functioning than the paid caregiver. Interventional studies showed that educating the caregivers improved their knowledge and reduced the burden, whereas the application of the continuous care model improved QOL, and the use of supportive and behavioral therapy contributed to the psychological adjustment of the caregivers [42]. Generally, coping strategies need to be learned and trained. Therefore, holding educational classes or preparing educational materials can be a useful solution, and caregivers can also take a day off from paid jobs [43]. While the majority of research has been conducted in middle-income countries focusing on reducing caregiver burden, it is critical to assess and address the practical issues that caregivers face, such as work-related responsibilities, financial difficulties, transportation to the dialysis unit, respite care, and the need to learn specific skills related to patients’ chronic illnesses [44]. Further, there is a need to strengthen professional nursing care rather than relying on family caregivers. In addition, Indonesia recently launched public health insurance, so they can increase professional nursing care for hemodialysis patients. 

As a limitation, first, the study site is a private hospital, where patients are from high- socioeconomic status. Therefore, the results of this study may not be generalized to other hemodialysis units in the country. Second, this study had a small sample size; therefore, further research in multiple centers with higher sample sizes is required. 

## 5. Conclusions

Caregivers’ QOL was found to be indirectly influenced by burden through anxiety and depression. Further evaluation of clinical significance and implications for caregiver’s lifestyle regarding anxiety and depression, which result in caregiver burden, is warranted. With this aim, it leads to a better understanding of the caregiver’s burden and may provide insight into which interventions will be most effective in reducing caregiver burden, anxiety, and depression, besides improving their QOL. 

It is critically important to assess the patients with chronic illnesses, especially those on dialysis, for the presence of psychiatric morbidity and initiate prompt treatment. This can improve compliance, long-term outcome, and prognosis in these patients. In order to enhance the QOL on hemodialysis patients and reduce the burden on their caregivers, governmental and non-governmental organizations need to expand special support groups that consist of patients, caregivers, as well as health staff, where they can share their knowledge, experiences, and ways to handle a crisis and improve treatment adherence.

## Figures and Tables

**Figure 1 ijerph-19-04544-f001:**
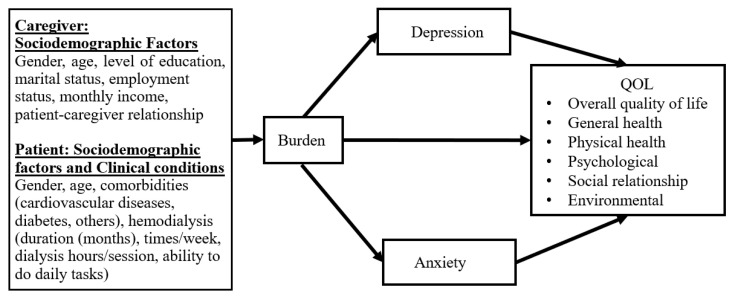
Conceptual framework of the association of sociodemographic and burden in this study.

**Table 1 ijerph-19-04544-t001:** Sociodemographic characteristics of patients and the caregiver.

Variables	Categories	*n* (%)
The Caregiver		
Sex	Male/Female	38 (36.5)/66 (63.5)
Age (Mean ± SD) years	(range 18–75)	44.4 ± 12.7
Level of education	Junior high school or lower	3 (2.9)
	Senior high school	26 (25.0)
	Diploma/Bachelor	68 (65.4)
	Master or above	7 (6.7)
Marital status	Single	27 (26.0)
	Married	71 (68.3)
	Widow	6 (5.8)
Employment status	Yes	56 (53.8)
	(employee/self-employee)	42 (40.4)/14 (13.5)
Monthly income *	Low	20 (19.2)
	Lower-Middle	22 (21.2)
	Upper-Middle	25 (24.0)
	High	37 (35.6)
Caregiver–Patient	Parent	12 (11.5)
relationship	Spouse	49 (47.1)
	Child	30 (28.8)
	Siblings	6 (5.8)
	Another relative	7 (6.7)
The Patient		
Sex	Male/Female	55 (52.9)/49 (47.1)
Age (Mean ± SD) years	(range 22–88)	56.9 ± 14.8
Comorbidities (chronic)	Hypertension	46 (44.2)
	Diabetes	33 (31.7)
	Cardiovascular	12 (11.5)
Hemodialysis duration (months)	(Range: 3–334) Mean ± SD	37.7 ± 46.1
	≤3 years	70 (67.3)
	≤6 years	22 (21.2)
	≤9 years	7 (6.7)
	More than 10 years	5 (4.8)
Hemodialysis time/week	1 time/week (4 h)	2 (1.9)
	2 times/week (92.9% = 5 h/session)	94 (90.4)
	3 times/week	7 (6.7)
	(4 h/session, 5 h/session)	(*n* = 4, *n* = 3)
	5 times/week (5 h/session)	1 (1.0)
Hemodialysis hours/session	3 h	1 (1.0)
	4 h	10 (9.6)
	4.5 h	2 (1.9)
	5 h	91 (87.5)
Ability to perform daily task	Independent	46 (44.2)
	Half-dependent	50 (48.1)
	Fully dependent	8 (7.7)

*n* = 104, * Low: less than Rp.1,000,000; Lower-Middle: Rp.1,000,000 to Rp.3,000,000; Upper-Middle: Rp.3,000,000 to Rp.5,000,000; High: more than Rp.5,000,000.

**Table 2 ijerph-19-04544-t002:** Questionnaire response scores *n* = 104.

Variables	Score Range	Mean ± SD	*n* (%)
ZBI	0–88	21.6 ± 13.7	
Little or no burden	0–20		55 (52.9)
Mild-to-moderate burden	21–40		37 (35.6)
Moderate-to-severe burden	41–60		11 (10.6)
Severe burden	61–88		1 (1.0)
HADS (Anxiety)	0–21	6.5 ± 4.7	
Normal	0–7		64 (61.5)
Borderline abnormal (borderline case)	8–10		14 (13.5)
Abnormal case	11–21		26 (25.0)
HADS (Depression)	0–21	5.6 ± 3.7	
Normal	0–7		73 (70.2)
Borderline abnormal (borderline case)	8–10		21 (20.2)
Abnormal case	11–21		10 (9.6)
WHOQOL-BREF			
Overall quality of life	1–5	3.7 ± 0.8	
General health	1–5	3.4 ± 0.9	
Physical health domain	13–94	68.1 ± 14.3	
Psychological domain	6–94	66.1 ± 15.3	
Social relationship domain	0–100	60.2 ± 16.4	
Environmental domain	0–4	63.2 ± 15.2	

**Table 3 ijerph-19-04544-t003:** Simple linear regression between sociodemographic and clinical-condition variables and burden.

Caregiver Characteristics	*β* [SE]	*p* Value	R^2^
Sex	0.78 [2.80]	0.780	0.001
Educational level	−0.81 [3.00]	0.790	0.001
Marital status	−2.10 [2.89]	0.470	0.005
Employment status	0.74 [2.70]	0.790	0.001
Income	−1.03 [2.75]	0.710	0.001
Relationship to the patient	0.94 [1.58]	0.560	0.003
Patients’ clinical conditions			
Hemodialysis duration (months)	2.74 [2.86]	0.340	0.009
Dependency level on daily activities	3.77 [2.69]	0.160	0.019

**Table 4 ijerph-19-04544-t004:** Stepwise multiple linear regression analysis for burden and sociodemographic and clinical factors in relation to QOL.

Independent Variables	Dependent Variable					
	Overall QOL	General Health	Physical Health	Psychological	Social Relationship	Environmental
Constant	3.794	3.627	78.364	75.022	66.077	85.318
	(0.734)	(0.778)	(12.459)	(12.711)	(14.257)	(12.735)
ZBI	−0.019 ***	−0.014 **	−0.342 ***	−0.489 ***	−0.414 ***	−0.480 ***
	(0.006)	(0.006)	(0.100)	(0.103)	(0.115)	(0.103)
Sex	0.121	-0.177	4.127	4.691	0.906	3.878
	(0.734)	(0.778)	(12.459)	(12.711)	(14.257)	(12.735)
Education	−0.067	0.027	−0.875	−1.356	1.872	−1.091
	(0.209)	(0.221)	(3.543)	(3.615)	(4.054)	(3.622)
Marital status	0.149	0.078	3.258	2.339	3.308	−2.321
	(0.191)	(0.202)	(3.235)	(3.300)	(3.702)	(3.307)
Occupation	0.124	−0.119	2.197	2.233	2.762	4.410
	(0.201)	(0.213)	(3.405)	(3.474)	(3.896)	(3.480)
Income	0.257	0.206	−4.037	−0.407	−0.189	−2.346
	(0.192)	(0.203)	(3.254)	(3.320)	(3.724)	(3.327)
Relationship	0.015	−0.042	−1.025	−2.165	−2.394	−2.574
	(0.103)	(0.109)	(1.748)	(1.783)	(2.000)	(1.787)
HD duration	−0.090	0.061	0.373	1.005	0.961	1.579
	(0.176)	(0.187)	(2.987)	(3.047)	(3.418)	(3.053)
Patient ability	−0.207	−0.163	−1.431	0.302	−1.887	−1.831
	(0.171)	(0.181)	(2.903)	(2.962)	(3.322)	(2.967)
R square	0.175	0.089	0.174	0.244	0.175	0.233
Adjusted R square	0.096	0.002	0.095	0.172	0.096	0.159
No observations	104	104	104	104	104	104
F statistic	2.217 **	1.021 *	2.204 **	3.373 ***	2.215 **	3.170 ***

Note: * *p <* 0.05; ** *p <* 0.01; *** *p <* 0.001.

**Table 5 ijerph-19-04544-t005:** Stepwise multiple linear regression analysis for HADS and sociodemographic and clinical factors in relation to QOL.

	Dependent Variable					
	Overall QOL	General Health	Physical Health	Psychological	Social Relationship	Environmental
Constant	3.929	3.901	80.442	80.696	72.933	76.313
	0.241	(0.131)	(1.931)	1.917	2.344	2.05
HADS	−0.075 ***	−0.054 ***	−1.297 ***	−1.528 ***	−1.335 ***	−1.372 ***
	(0.010)	(0.011)	(0.166)	(0.164)	(0.200)	(0.176)
Income	0.285					
	(0.132)					
R square	0.400	0.185	0.376	0.459	0.304	0.374
Adjusted R square	0.388	0.177	0.37	0.454	0.297	0.368
Durbin–Watson ratio	2.060	1.966	1.626	1.971	2.032	1.63
No observations	104	104	104	104	104	104
F statistic	33.691 ***	23.180 ***	61.376 ***	86.499 ***	44.525 ***	60.926 ***

Note: *** *p <* 0.001.

**Table 6 ijerph-19-04544-t006:** Path analysis: model fitness of all QOL domains (*n* = 104).

	Burden-HADS			HADS-QOL			Model Fitness					
	*β*		R^2^	*β*		R^2^	χ^2^	df	*p* Value	AGFI	CFI	RMSEA
Overall QOL	0.7	***	0.45	−0.61	***	0.37	1.654	1	0.198	0.937	0.994	0.344
Overall health				−0.43	***	0.19	0.505	1	0.477	0.981	1.000	0.000
Physical				−0.59	***	0.35	1.204	1	0.273	0.954	0.998	0.044
Psychological				−0.65	***	0.42	0.000	1	0.988	1.000	1.000	0.000
Social relationship				−0.54	***	0.29	0.000	1	0.988	1.000	1.000	0.000
Environmental				−0.59	***	0.35	0.252	1	0.616	0.99	1.000	0.000

*** *p* < 0.001, HADS: Hospital Anxiety and Depression Scale, which measures anxiety and depression.

## Data Availability

On request, the data used in this research can be obtained from the corresponding author.

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
