# Peer review of "Assessing Burden, Anxiety, Depression, and Quality of Life among Caregivers of Hemodialysis Patients in Indonesia: A Cross-Sectional Study"

_ijerph, 2022, doi:10.3390/ijerph19084544_

Round 1

Reviewer 1 Report

Thank you very much for the opportunity to review the manuscript entitled: "Assessing burden, anxiety, depression, and quality of life among caregivers of hemodialysis patients in Indonesia: A cross-sectional study". This is a very interesting research topic. The following are some considerations:

TITLE

Adequate.

ABSTRACT

Adequate. Maintains a logical structure and exposes the main parts of the study.

KEY WORDS

Adequate.

INTRODUCTION

Adequate. I suggest using the third personal singular instead of the first person plural. This aspect should be revised throughout the manuscript to provide the study with greater objectivity in writing.

METHOD

Line 176. There may be an extra space before "Next".

When describing the instrument, an example item and Cronbach's Alpha of the scales should be given.

RESULTS

Adequate.

DISCUSSION

More emphasis should be placed on future lines of research.

CONCLUSION

Adequate.

REFERENCES

Adequate.

Thank you for your attention. 

Author Response

Response to the reviewers

Reviewer 1

Thank you for your kind observation and valuable recommendation. We revised based on your suggestions. Please see our responses below.

Comments

1) TITLE: Adequate.

Response

Thank you.

2) ABSTRACT: Adequate. Maintains a logical structure and exposes the main parts of the study.

Response

Thank you for your suggestion. We added a logical structure accordingly.

3) KEY WORDS: Adequate.

Response

Thank you.

4) INTRODUCTION: Adequate. I suggest using the third personal singular instead of the first person plural. This aspect should be revised throughout the manuscript to provide the study with greater objectivity in writing.

Response

We revised accordingly.

5) METHOD: Line 176. There may be an extra space before "Next".

Response

We have removed the space before "Next".

6) When describing the instrument, an example item and Cronbach's Alpha of the scales should be given.

Response

We have given the Cronbach's Alpha of the HADS and WHOQOL-BREF scales on the line no 152 to 153 and 164 to 166.

7) RESULTS: Adequate.

Response

Thank you.

8) DISCUSSION: More emphasis should be placed on future lines of research.

Response

We added the below sentence in the last part of the “Discussion” part.

While the majority of research has been conducted in middle-income countries focusing on reducing caregiver burden, it is critical to assess and address the practical issues that caregivers face, such as work-related responsibilities, financial difficulties, transportation to the dialysis unit, respite care, and the need to learn specific skills related to patients' chronic illnesses. Further, it’s needed to strengthen professional nursing care rather relying on family caregivers. Besides, Indonesia recently launched public health insurance, so they can increase professional nursing care for heamodialysis patients.

Reviewer 2 Report

Caregivers for hemodialysis patients suffer anxiety and depression and poor quality of life to a significant degree. The purpose of the study is to assess the emotional burden and quality of life of the caregivers and their relation to socioeconomic factors and the patient's clinical conditions. The authors hypothesize that caregivers' socioeconomic factors patient's clinical conditions affect the level of burden and burden affects the quality of life.

I'm not sure what the purpose of the study is. Obviously, health workers can experience emotional distress and poor quality of life. I don't see what the findings mean. The authors refer to future studies but this study should have looked into the negative effects on patient care and how the health care workers can be treated psychologically. In regard to the statistics, I think a mixed model would be more appropriate and I would suggest that a statistician review the statistics.

Author Response

Reviewer 2

Thank you for your review. We revised based on your suggestions. Please see our responses below.

Comments

I'm not sure what the purpose of the study is. Obviously, health workers can experience emotional distress and poor quality of life. I don't see what the findings mean. The authors refer to future studies but this study should have looked into the negative effects on patient care and how the health care workers can be treated psychologically. In regard to the statistics, I think a mixed model would be more appropriate and I would suggest that a statistician review the statistics.

Response

We clarified the below.

(1) Our focus is on family caregivers, not professional caregivers such as nurses. Therefore, in the abstract, we added: “family” in front of “caregivers”.

(2) Our study purpose is described in the last part of “The introduction” part. We also added “family” in front of “caregiver” to clarify the purpose.

The purposes of this study were (1) to assess the level of burden, anxiety, depression, and QOL among family caregivers of patients receiving hemodialysis in Indonesia, (2) to investigate the influence of family caregivers’ sociodemographic factors and patients’ clinical conditions on the level of burden, and (3) to investigate how burden affects anxiety/depression and QOL.

(3) Our study conclusion is that “family caregivers’ QOL was found indirectly influenced by burden through anxiety/depression”. We clarified the sentence in our conclusion part.

We have added future implications regarding the sentence in our discussion part.

(4) Two statisticians (2nd and 4th authors) conducted this statistical analysis. A mixed-effect model is used to address the multi-level data or longitudinal data. Our data was collected at a single time point in a single hospital. Therefore, there is no inter-correlation among participants; hence we do not need to do the mixed model.

Reviewer 3 Report

In general, the paper is sound. The English is pretty good but would benefit from review in place.

In section 2.2, it seems as though the assessments were completed in the hospital, so it would be worth stating this. 

In section 3.5, does the path analysis alter the hypothesis set out in Fig. 1? A revised figure might be helpful to illustrate this. I'd imagine the the direct arrow from burden to carer QoL might disappear.

In the discussion, the authors correctly point out that their sample was relatively prosperous and receiving treatment in a private hospital, which might contribute to the relatively low levels of burden recorded. 

Were there any differences in burden and other variables between different groups of caregivers, especially children compared to spouses? 

Finally, it is perhaps not surprising that mental health (anxiety, low mood) were the main factors influencing carer burden and QoL. Does the study tell us anything novel and are there any implications for clinical practice as to how caregivers of haemodialysis papers might be better supported? At present, this seems to be missing.

Author Response

Reviewer 3

Thank you for your kind observation and valuable recommendation. We revised based on your suggestions. Please see our responses below.

Comments

1) In section 2.2, it seems as though the assessments were completed in the hospital, so it would be worth stating this.

Response

We added the underline part in the first sentence.

A convenience sampling procedure was used for this study in the selected above-mentioned hospital.

2) In section 3.5, does the path analysis alter the hypothesis set out in Fig. 1? A revised figure might be helpful to illustrate this. I'd imagine the direct arrow from burden to carer QoL might disappear.

Response

Thank you for the suggestion. As result, a direct line from the burden to QOL was disappeared. However, Fig. 1 shows the hypothesis. We analyze the data based on Fig. 1, therefore we cannot change this framework. However, we added path analysis figures in a supplemental file (On this figure, direct lines from burden to QOL do not exist.).

3) In the discussion, the authors correctly point out that their sample was relatively prosperous and receiving treatment in a private hospital, which might contribute to the relatively low levels of burden recorded.

Response

Thank you for your compliment.

4) Were there any differences in burden and other variables between different groups of caregivers, especially children compared to spouses?

Response

Thank you for this concern. The “Relationship to the patient” indicates parent, spouse, child, siblings, another relative, caregiver. Based on your suggestion, we conducted a t-test on ZBI, HADS (Anxiety and Depression), and QOL for each domain (please see the table below). Actually, HADS (Anxiety & Depression) and WHOQOL (4 domains) showed statistically significant differences between a child and a spouse, but no statistically significant difference in ZBI (burden).

As we hypothesized that sociodemographic factors affect burden, not anxiety, depression, and QOL, we will not add or change the results of this manuscript. In addition, Table 3 partially showed the linear regression result between the “relationship of the patient” and “burden” with no significance.

However, thank you for this suggestion.

5) Finally, it is perhaps not surprising that mental health (anxiety, low mood) were the main factors influencing carer burden and QoL. Does the study tell us anything novel and are there any implications for clinical practice as to how caregivers of haemodialysis papers might be better supported? At present, this seems to be missing.

Response

Thank you for your supporting comment. We added our novelty at the beginning of the discussion part as below.

To our knowledge, this is the first exploratory study on burden, anxiety, depression, and QOL among family caregivers of hemodialysis patients in the Indonesian context.

Further, we have given an explanation of clinical practice implications in the last second paragraph of the discussion section.

Round 2

Reviewer 2 Report

I feel the author need to explain why burden increased anxiety and depression but not QOL. If workers have increased anxiety and depression shouldn't that lead to a decrease in QOL? Also, a mixed model analysis with subjects as a random effect would be more appropriate.

Author Response

Response to the reviewer

Reviewer 2

Thank you for your further review. We added interpretations of the results. Please see our responses below.

Comments

I feel the author need to explain why burden increased anxiety and depression but not QOL. If workers have increased anxiety and depression shouldn't that lead to a decrease in QOL?

Response

In the abstract and the result sections, we explained that "the burden influenced QOL via depression/anxiety," and there was no effect of a burden on QOL. This means that our hypothesis was not supported.

We clarified the below (the underlined part is added).

1) We added the interpretation of the results in lines 239 to 240.

“The results showed that burden (Table 4) and HADS (Table 5) were significantly, negatively associated with all domains of QOL (all, p <0.001), which indicates when the burden or HADS (anxiety and/or depression) became severe, QOL (all domains) decreases.

2) We added the interpretation in the discussion part.

Page 10, Line 302-304

“We assumed that WHOQOL includes items such as physical pain, self-esteem, activities of daily living, personal relations, and home environment, which might not be affected by the caregiving burden.”

Comment

A mixed model analysis with subjects as a random effect would be more appropriate.

Response

As we explained in the previous response, the mixed effect model is used to consider the intra-cluster of the data when the data has a hierarchy. If we collected the data multiple times from the same subjects, it would be appropriate to include a random effect of subjects as explained in a previously published paper in IJERPH (https://www.mdpi.com/1660-4601/18/21/11234).

However, we collected the data only at once from each subject, so there is no random effect that we need to consider.

Would you give us a clear scientific rationale if you suggested utilizing it? Even if their data was only acquired at a single time point, how could we account for the subject's random effect?

For this reason, we believe the mixed model is not necessary for this study.
